# GALERKIN MEETS LAPLACE:
# FAST UNCERTAINTY ESTIMATION IN NEURAL PDES

**Christian Jimenez Beltran**[1]**, Antonio Vergari**[2]*****, Aretha L. Teckentrup**[1]*****,
**Konstantinos C. Zygalakis**[1]* *

[1]School of Mathematics and Maxwell Institute for Mathematical Sciences, [2] School of Informatics
University of Edinburgh
{s2113174,avergari,a.teckentrup,k.zygalakis}@ed.ac.uk

## ABSTRACT

The solution of partial differential equations (PDEs) by deep neural networks trained to satisfy the differential operator has become increasingly popular. While these approaches can lead to very accurate approximations, they tend to be over-confident and fail to capture the uncertainty around the approximation. In this work, we propose a Bayesian treatment to the deep Galerkin method (Sirignano & Spiliopoulos, 2018), a popular neural approach for solving parametric PDEs. In particular, we reinterpret the deep Galerkin method as the maximum a posteriori estimator corresponding to a likelihood term over a fictitious dataset, leading thus to a natural definition of a posterior. Then, we propose to model such posterior via the Laplace approximation, a fast approximation that allows us to capture meaningful uncertainty in out of domain interpolation of the PDE solution and in low data regimes with little overhead, as shown in our preliminary experiments.

## 1 INTRODUCTION

Partial differential equation (PDE) models appear in numerous areas of science and engineering, including geophysics, climate modelling and medical imaging to name a few (De Marsily, 1986; Isaacson et al., 2004; Schneider et al., 2017). In the last few years, deep learning approaches that use neural networks to approximate the solution of PDEs have shown remarkable success in terms of approximating certain classes of high-dimensional PDEs (Raissi et al., 2019; Kelshaw et al., 2022), thus providing fast alternatives to classical numerical methods such as finite differences, elements, volumes and spectral methods (Larsson & Thomée, 2003).

One of the first deep learning approaches designed for solving PDEs is the physics-informed neural networks (PINNs) (Raissi et al., 2019). PINNs define a suitable loss function that takes explicitly into account the PDE in question over specific collocation points, as well as over specific initial and boundary data. To do so, PINNs require fixed collocation points at training time, an aspect that can become problematic for high-dimensional PDEs. In addition, in many applications, one is interested in solving the PDE for a range of parameters, which is something that cannot be addressed by PINNs. The deep Galerkin method (DGM) (Sirignano & Spiliopoulos, 2018) can deal with these two issues by averaging oversampled collocation points and initial and boundary data during training and adding the PDE parameters as additional inputs to the neural network.

While these neural approaches can scale PDE solving to high dimensions, they still require large amounts of data to be trained and tend to be highly overconfident even when their approximations are far from the ground truth. This is especially problematic when they are used as surrogate solutions on collocations unseen during training (out-of-sample) and when trying to perform statistical analysis of real-world differential equation models, as it is important to have calibrated estimates of uncertainty with regards to our numerical approximations (Hennig et al., 2015; Conrad et al., 2017).

In this work, we address this issue for DGMs. We start by reinterpreting the DGM as the maximum a posteriori estimator corresponding to a likelihood term over a fictitious data set, leading thus to a natural Bayesian treatment. Then, we propose a simple and effective solution: approximating the

---

*Shared supervision.

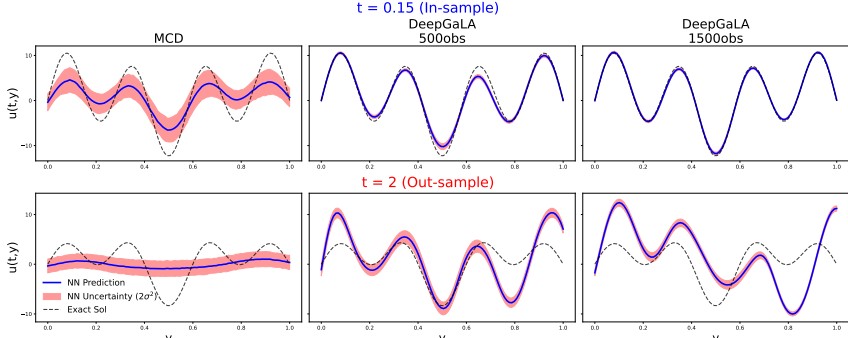

Figure 1: **Our DEEPGALA extends the DGM to provide meaningful uncertainty on out-of-sample** input locations (bottom) for the PDE solution of (PDE1) for $\theta = 0.001$ and when trained over 1000 (second column) and 3000 (third column) samples respectively. Monte Carlo Dropout (MCD) struggles to approximate the solution even in-sample.

corresponding posterior via Laplace's method. This allows us to deliver fast uncertainty estimation over the PDEs that can be meaningful when training data is scarce or we evaluate our surrogate solution on out-of-sample collocations (see Figure 1 and Figure 2).

## 2 THE DEEP GALERKIN METHOD

We consider a general parametric PDE (see e.g. Deveney et al. (2023)), given by

$$\mathcal{A}(x, u(x;\theta); \theta_A) = h(x;\theta_h), \quad u(\partial x;\theta) = b(\partial x;\theta_b), \quad x \in \Omega, \quad \partial x \in \partial\Omega, \quad \theta \in \Theta \quad (1)$$

where the domain $\Omega \subset \mathbb{R}^d$ is the input space with boundary $\partial\Omega$, and $\Theta \subset \mathbb{R}^p$ is the parameter space for $\theta = (\theta_A, \theta_h, \theta_b)$. The differential operator $\mathcal{A}$ is parameterized by $\theta_A$, whereas the functions $h$ and $b$ are parameterized by $\theta_h$ and $\theta_b$, respectively. We denote by $u(x;\theta)$ the solution of the parametric PDE. Note that $\mathcal{A}$ could be non-linear, and that time could be a component of the input $x$. We want to use a neural network (NN) to approximate $u(x;\theta)$. To this end, consider a NN $f_{\mathrm{W}}$ with $L$ layers and parameterized by $\mathrm{W} = \{\mathrm{W}_1, \ldots, \mathrm{W}_L\}$.[1] The NN takes as inputs a point $x \in \overline{\Omega}$ in the closure of the input domain and a parameter value $\theta \in \Theta$, and it is trained to approximate the solution of the PDE, $f_{\mathrm{W}}(x;\theta) \approx u(x;\theta)$.

The deep Galerkin method (DGM) (Sirignano & Spiliopoulos, 2018) uses a mesh-free strategy: At each iteration of the training step, $K$ collocations for $x_i, \partial x_i$, and $\theta_i$ are sampled from densities $\pi^p, \pi^b$, and $\pi^\theta$, respectively. Then, the NN $f_{\mathrm{W}}$ is trained to minimize the following loss:

$$\frac{1}{K}\sum\nolimits_{i=1}^{K} \ell(D_i; \mathrm{W}) = \frac{1}{K}\sum\nolimits_{i=1}^{K} \left(\ell_h(D_{h,i}; \mathrm{W}) + \ell_b(D_{b,i}; \mathrm{W})\right), \quad (2)$$

where $\mathcal{D} = \{D_i\}_{i=1}^{K} = \{D_{h,i}\}_{i=1}^{K} \cup \{D_{b,i}\}_{i=1}^{K}$, and $D_{h,i} = \{x_i, \theta_i\}$ and $D_{b,i} = \{\partial x_i, \theta_i\}$. Here, $\ell_h(D_{h,i}; \mathrm{W}) = (\mathcal{A}(x_i, f_{\mathrm{W}}(x_i, \theta_i); \theta_{A,i}) - h(x_i; \theta_{h,i}))^2$ measures the error in the approximation of the differential operator and $\ell_b(D_{b,i}; \mathrm{W}) = (f_{\mathrm{W}}(\partial x_i, \theta_i) - b(\partial x_i; \theta_{b,i}))^2$ measures the error in the boundary conditions. Optimization is carried out by gradient-based approaches, e.g., SGD, possibly increasing the number of samples $K$ and the number of iterations. In this work, we followed a two-step training as suggested by He et al. (2020). Although conceptually simple, there are no convergence guarantees for DGM, and the error in predictions made by the NN $f_W$ could be significant. In applications where the NN is then used in a computational pipeline, such as using NNs to approximate the likelihood in an inverse problem (Deveney et al., 2023), it becomes crucial to quantify the uncertainty in the predictions of the NN to avoid overconfident and biased inference.

---

[1]We consider the biases to be included in each $\mathrm{W}_l$. Note that other non-sequential architectures are possible.

## 3 DEEPGALA: LAPLACE APPROXIMATION FOR BAYESIAN DGM

In this section, we explain how to extend the DGM to allow for uncertainty quantification in its predictions, using Bayesian theory and the Laplace approximation. Uncertainty quantification has been thoroughly investigated for NNs as function approximators (Wilson & Izmailov, 2020; Wilson, 2020) and more recently also for neural PDE solvers such as PINNS and DeepOnets (Psaros et al., 2023). To the best of our knowledge, this has not been done for neural parametric PDE solvers such as DGM yet. Moreover, our approach emphasises more precisely assessing the uncertainty resulting from limited sample numbers, training methods, and NN hyperparameters.

### 3.1 BAYESIAN RE-INTERPRETATION OF DGM

In a Bayesian version of DGM, we are interested not in a single parameter configuration W, but in its posterior distribution $p(\text{W} \mid \mathcal{D})$. To this end, we first interpret the data $\mathcal{D}$ to be the fictious dataset built by $K$ collocations points sampled from $\pi^p, \pi^b$, and $\pi^\theta$. Then, we interpret the loss in equation 2 as a negative log-likelihood by considering Gaussian noise around the fictitious data observed at the collocation points. By doing this, the likelihood $p(\mathcal{D}|\text{W})$ characterises how well a choice of weights $W$ approximates the PDE solution $u(x; \theta)$, and is given by

$$p(\mathcal{D}|\text{W}) = p(D_h|\text{W})p(D_b|\text{W}) = \frac{1}{(2\pi\sigma^2)^{2K}} \prod_{i=1}^{K} \exp\left(-\frac{\ell_h(D_{h,i}; \text{W}) + \ell_b(D_{b,i}; \text{W})}{2\sigma^2}\right), \quad (3)$$

where we assume that the likelihoods of the collocations $D_h$ and $D_b$ are conditionally independent given W. We further introduce a Gaussian prior distribution $p(W)$. By Bayes' rule, we then have the posterior distribution $p(W|\mathcal{D}) \propto p(\mathcal{D}|\text{W})p(W)$ on the weights $W$ given the fictitious training data $\mathcal{D}$. Note that although not explicitly indicated, the posterior $p(W|\mathcal{D})$ is also conditioned on $\mathcal{A}, h$ and $b$ from equation 1, since these are used to define the likelihood. While intractable in general, $p(W|\mathcal{D})$ is amenable to a fast and and effective approximation, as discussed next.

### 3.2 A LAPLACE APPROXIMATION OF BAYESIAN DGM

We propose a Laplace Approximation (LA) of the posterior of our Bayesian DGM, which we call DEEPGALA– Deep Galerkin via Laplace. The LA is a well-known method for approximating intractable posterior distributions as a Gaussian distribution centred on the maximum a posteriori (MAP) solution $W_{\text{MAP}}$, i.e.,

$$\arg\min_{W} \frac{1}{2\sigma^2} \sum_{i=1}^{K} \ell(D_i; \text{W}) + r(W), \quad (4)$$

where the negative log-prior $r(W) = -\log p(W)$ is an L2 regularizer (weight decay) in our experiments. Despite its simplicity (assuming unimodality of the posterior), the LA has been demonstrated to be a solid alternative to more complex approximations such as MCMC and VI on a number of uncertainty quantification tasks for Bayesian deep learning (Daxberger* et al., 2021). To compute the LA, the first step is to find the minimizer $W_{\text{MAP}}$ of equation 4, which can be achieved by training $f_{\text{W}}$ via gradient-based optimization. The second step is to fit the local Gaussian distribution, namely $\mathcal{N}(W; W_{\text{MAP}}, \Lambda)$ where $\Lambda^{-1}$ is the Hessian of the negative log-posterior evaluated at $W_{\text{MAP}}$. In our case, $r(W)$ is a weight decay regularizer, which corresponds to a Gaussian prior distribution $p(W) = \mathcal{N}(W; 0, \gamma^2 I)$ (Daxberger* et al., 2021), $\Lambda^{-1}$ taking the form

$$\Lambda^{-1} = -\frac{1}{2\sigma^2} \sum_{i=1}^{K} \nabla_W^2 \log p(D_i|W)|_{W_{\text{MAP}}} - \gamma^{-2} I. \quad (5)$$

Exactly computing $\Lambda^{-1}$ above can be computationally demanding, as W in modern NNs can be very large (Nilsen et al., 2019). To this end, we employ a fast approximation of the covariance in DEEP-GALA, by leveraging a number of heuristics from the modern Bayesian deep learning literature. First, we consider only the weights $\text{W}_L$ of the last layer of $f$, which is generally enough to deliver good uncertainty estimates as noted in Sharma et al. (2023). [2] Second, we subsample a subset $K'$ of all the training points seen by $f$ during training. Third, we either compute the full Hessian or approximate the Hessian by its diagonal (Pearlmutter, 1994). Appendix A details the whole process.

---

[2]This also allows us to compute the posterior predictive in closed-form, see Appendix A.1 for details.

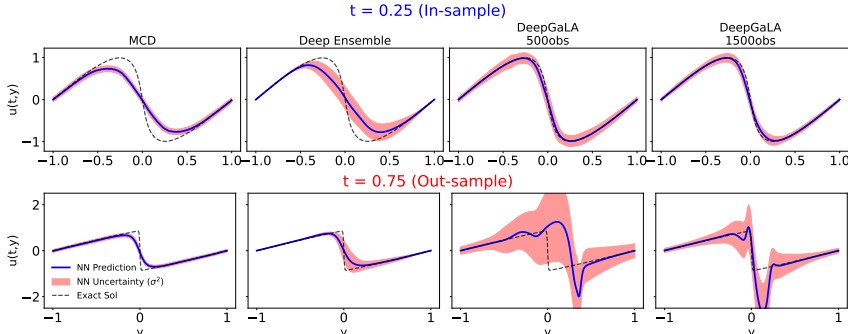

Figure 2: **DEEPGALA provides meaningful uncertainty on out-of-sample** input locations (bottom) for the Burger's solution of equation PDE2 for $\theta = 0.01/\pi$ and when trained over 500 (third column) and 1500 (fourth) samples. Monte Carlo Dropout (MCD) and Deep Ensembles struggled to approximate the solution of equation PDE2.

## 4 EXPERIMENTS AND DISCUSSION

We aim to answer the following research questions: **RQ1)** does DEEPGALA capture meaningful uncertainty over out-of-sample inputs $x$? **RQ2)** does increasing training size properly reduce the epistemic uncertainty highlighted by DEEPGALA?

**PDEs.** To this end, we consider two PDEs defined over $x = (y, t)$: A heat equation with external heat source:

$$\frac{\partial u}{\partial t} = \theta \frac{\partial^2 u}{\partial y^2} + \sin(5\pi y), \quad t > 0, \quad \theta > 0 \quad \text{and} \quad y \in [0, 1], \tag{PDE1}$$

with conditions $u(y, 0) = 4\sin(3\pi y) + 9\sin(7\pi y)$, and $u(0, t) = u(1, t) = 0$; and Burgers Equation

$$\frac{\partial u}{\partial t} + u\frac{\partial u}{\partial y} - \theta\frac{\partial^2 u}{\partial y^2} = 0, \quad t > 0, \quad \theta > 0 \quad \text{and} \quad y \in [-1, 1], \tag{PDE2}$$

with conditions $u(y, 0) = -\sin(\pi y)$, and $u(-1, t) = u(1, t) = 0$. PDE1 admits analytic solutions, allowing us to directly compare the performance of DEEPGALA both in terms of accuracy and uncertainty. For PDE2, we use the results obtained by the work of Raissi et al. (2019) for $\theta = 0.01/\pi$ as a baseline.

The experiments were carried out in a machine with 12th Gen Intel Core i7-1265Ux12, 32 GB of RAM and Ubuntu 22.04 operating system.

**NNs models.** For both PDEs we use a feed-forward NN with three layers and 40 neurons using hyperbolic tangent non-linearities, trained with Adam for 1,200 epochs, a learning rate of 0.01 and regularizer, $\gamma$, of 0.0015 for PDE1 and 0.0001 for PDE2, followed by the L-BFGS optimizer in a two-step optimisation, as recommended by He et al. (2020). We show results for the diagonal approximation in Appendix B. The mean time to fit and call the full Hessian LA for PDE1 and PDE2 was 0.081 and 0.003, and 0.077 and 0.0044 seconds respectively.

**Alternative UQ baselines.** We test our approach against Deep Ensemble (DE) (Lakshminarayanan et al., 2017b) and Monte Carlo Dropout (MCD) (Lakshminarayanan et al., 2017a). In order to calculate the mean and variance for the MCD, 1,000 samples were used, with the dropout layer set to $p = 0.05$ for PDE1 and $p = 0.1$ for PDE2, respectively. For the PDE2, the NN architecture stays the same, however for the PDE1, five hidden layers comprising forty neurons were employed. The reason for the size modification was that smaller NN were not producing satisfactory results for this specific PDE. Five NN's of three layers and forty neurons were trained to approximate the PDE2 using the DE method in accordance with Lakshminarayanan et al. (2017a). Nevertheless, obtaining satisfactory results for the PDE1 was surprisingly challenging; this difficulty may have arisen from the PDE's initial condition. That's the reason why we don't present the results utilizing this baselines for the PDE1.

**RQ1) In- vs out-of-sample.** For both PDEs, we provide DEEPGALA only a portion of the input at training time (in-sample), expecting it to extrapolate poorly – but delivering good uncertainty – on out-of-sample inputs. In particular, we draw our in-sample collocations from the following intervals: $y_i \sim \mathcal{U}[0,1]$, $t_i \sim \mathcal{U}[0,1]$, $\theta_i \sim \mathcal{U}[0.0001, 0.05]$ for PDE1 and $y_i \sim \mathcal{U}[-1,1]$, $t_i \sim \mathcal{U}[0, 0.5]$, $\theta_i \sim \mathcal{U}[0.0001, 0.05]$ for PDE2, where $\mathcal{U}$ denotes the uniform distribution. The top (in-sample) and bottom (out-of-sample) sides of Figure 1 and Figure 2 show two uncertainty surfaces representing 95% and 68% confidence intervals around the MAP estimate for DEEPGALA for PDE1 and PDE2 respectively. Reassuringly, these uncertainties increase when the model makes more mistakes, highlighting how DEEPGALA can be used to indicate that the NN are "aware" that they may be forecasting incorrectly. In contrast, the uncertainty exhibits no specific behaviour when examined both in- and out-of-sample for the methods MCD and DE, nor do they provide good approximations at the mean.

**RQ2) training size effect.** We train DEEPGALA over sample sizes of 1,000, and 3,000 for PDE1 and 500, and 1,500 for PDE2, plotting the results in the top and bottom halves of Figure 1 and Figure 2. All models trained with more samples outperform the other (both for in-sample and out-sample values of $t$). Nonetheless, the exact solution is within the uncertainty range of the models for this particular value of $t$. Again, DEEPGALA is shown to be an effective way to estimate the epistemic uncertainty of NNs for PDE solving, which can be useful in quantifying how much data can improve performance. As such, we plan to use the uncertainty DEEPGALA provides in an active learning loop to select what are the most promising collocation to improve accuracy of the PDE solution.

## 5 CONCLUSION AND DISCUSSION

We proposed DEEPGALA as a fast and effective way to estimate uncertainty in deep neural PDE solvers, and tested it on preliminary benchmarks with encouraging results. We plan to evaluate DEEPGALA to more complex PDEs, extending it with recent advancements in Bayesian deep learning (Wilson, 2020) and comparing to other Bayesian PDE solvers such as the ones presented in Psaros et al. (2023), which however are limited to non-parametric PDEs. We hope that scaling uncertainty estimation can lead to a renewed interest for efficient probabilistic numerics for complex neural PDEs in the real-world.

## ACKNOWLEDGEMENTS

The Maxwell Institute of Mathematical Sciences and the University of Edinburgh's School of Mathematics provided funding for this research. AV was supported by the "UNREAL: Unified Reasoning Layer for Trustworthy ML" project (EP/Y023838/1) selected by the ERC and funded by UKRI EPSRC

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

## A  EFFICIENT COMPUTATION OF THE LAPLACE APPROXIMATION

Following Sharma et al. (2023), we consider partially stochastic networks and introduce uncertainty only in $W_L$, the parameters in the last layer of $f_W$. The last layer, involving only an affine transformation, can be represented as $f_W(x, \theta) = W_L \cdot [1, \phi(y_{L-1}(x, \theta))]$, where $W_L$ is the vector of weights and biases, and $\phi(y_{L-1}(x, \theta))$ is the vector of outputs of the penultimate layer for input $(x, \theta)$. Let's just focus on calculating the Hessian for one input $z_i = (x_i, \theta_i)$ and compute $\nabla^2_{W_L} \ell(z_i; W)$, where we will use the abbreviation $l_i$ and $f_{i,W}$ to denote $\ell(z_i; W)$ and $f_W(z_i)$. Therefore,

$$
\nabla^2_{W_L} l_i = \nabla_W
\begin{bmatrix}
\frac{\partial l_i}{\partial f_{i,W}} \frac{\partial f_{i,W}}{\partial w_L^0} \\
\frac{\partial l_i}{\partial f_{i,W}} \frac{\partial f_{i,W}}{\partial w_L^1} \\
\vdots \\
\frac{\partial l_i}{\partial f_{i,W}} \frac{\partial f_{i,W}}{\partial w_L^K}
\end{bmatrix}
=
\begin{bmatrix}
\frac{\partial^2 l_i}{\partial f_{i,W}^2} \frac{\partial f_{i,W}}{\partial w_L^0} \frac{\partial f_{i,W}}{\partial w_L^0} & \cdots & \frac{\partial^2 l_i}{\partial f_{i,W}^2} \frac{\partial f_{i,W}}{\partial w_L^0} \frac{\partial f_{i,W}}{\partial w_L^K} \\
\vdots & \ddots & \vdots \\
\frac{\partial^2 l_i}{\partial f_{i,W}^2} \frac{\partial f_{i,W}}{\partial w_L^K} \frac{\partial f_{i,W}}{\partial w_L^0} & \cdots & \frac{\partial^2 l_i}{\partial f_{i,W}^2} \frac{\partial f_{i,W}}{\partial w_L^K} \frac{\partial f_{i,W}}{\partial w_L^K}
\end{bmatrix}
+
$$

$$
\begin{bmatrix}
\frac{\partial l_i}{\partial f_{i,W}} \frac{\partial^2 f_{i,W}}{\partial w_L^{0^2}} & \cdots & \frac{\partial l_i}{\partial f_{i,W}} \frac{\partial^2 f_{i,W}}{\partial w_L^0 \partial w_L^K} \\
\vdots & \ddots & \vdots \\
\frac{\partial l_i}{\partial f_{i,W}} \frac{\partial^2 f_{i,W}}{\partial w_L^K \partial w_L^0} & \cdots & \frac{\partial l_i}{\partial f_{i,W}} \frac{\partial^2 f_{i,W}}{\partial w_L^{K^2}}
\end{bmatrix}. \quad (6)
$$

In the special case of a last layer with only an affine transformation, the second term in the previous equation becomes zero. Additionally, the terms $\frac{\partial f_{i,W}}{\partial w_L^k} = \phi_k(y_{L-1}(z_i))$ and $\frac{\partial f_{i,W}}{\partial w_L^0} = 1$. Consequently, we can reformulate the previous equation as:

$$
\nabla^2_{W_L} l_i = \mathbf{J}_{W_L}(f_{i,W}) \frac{\partial^2 l_i}{\partial f_{i,W}^2} \mathbf{J}_{W_L}(f_{i,W})^T =
$$

$$
= \frac{\partial^2 l_i}{\partial f_{i,W}^2}
\begin{bmatrix}
1 & \phi_1(y_{L-1}(z_i)) & \cdots & \phi_K(y_{L-1}(z_i)) \\
\phi_1(y_{L-1}(z_i)) & \phi_1(y_{L-1}(z_i))^2 & \cdots & \phi_K(y_{L-1}(z_i))\phi_1(y_{L-1}(z_i)) \\
\vdots & \vdots & \ddots & \vdots \\
\phi_K(y_{L-1}(z_i)) & \phi_1(y_{L-1}(z_i))\phi_K(y_{L-1}(z_i)) & \cdots & \phi_K(y_{L-1}(z_i))^2
\end{bmatrix}.
$$
$$(7)$$

Another observation is that the last equation is akin to the Gauss-Newton Matrix (GNN), up to a constant $G(W) = \frac{1}{N} \sum_{n=1}^N \mathbf{J}_W(f_W(z_n))^T \nabla^2_f l(f_W(z_n)) \mathbf{J}_W(f_W(z_n))$. The GNN is a first-order Taylor approximation of the loss function. We can compute the equation 5 by using the next equation:

$$
\Lambda^{-1} = -\gamma^{-2} I - \frac{1}{2\sigma^2} \sum_{i=1}^K (\mathbf{J}_{W_L}(f_W(z_i)) \nabla^2_f \ell(D_{h,i}; W) \mathbf{J}_{W_L}(f_W(z_i))^T +
$$

$$
\mathbf{J}_{W_L}(f_W(\partial z_i)) \nabla^2_f \ell(D_{b,i}; W) \mathbf{J}_{W_L}(f_W(\partial z_i))^T), \quad (8)
$$

where $\mathbf{J}_{W_L}(f_W(z_i)) = [\frac{\partial f_W(z_i)}{\partial w_L^0}, ..., \frac{\partial f_W(z_i)}{\partial w_L^c}]^T$. Note that equation 8 represents a full Hessian; for a big enough NN, we can approximate even further to obtain a diagonal approximation by using the Hadamard product. Thus, the Diagonal approximation of the Hessian is computed in the following way:

$$\Lambda^{-1} = -\gamma^{-2}I - \frac{1}{2\sigma^2}\sum_{i=1}^{K}(\mathbf{J}_{W_L}\left(f_W(z_i)\right) \odot \nabla_{\hat{f}}^2\ell(D_{h,i};\mathrm{W}) \odot \mathbf{J}_{W_L}\left(f_W(z_i)\right)^T +$$
$$\mathbf{J}_{W_L}\left(f_W(\partial z_i)\right) \odot \nabla_{\hat{f}}^2\ell(D_{b,i};\mathrm{W}) \odot \mathbf{J}_{W_L}\left(f_W(\partial z_i)\right)^T). \quad (9)$$

## A.1 POSTERIOR PREDICTIVE

Finally, in order to perform predictions, one needs to compute the posterior predictive distribution. This is done in the following way. Given our focus on the last layer of $f_W$, linearity in the weights $W$ is observed. Therefore, for a test input $\hat{z}_i$ of the neural network, we obtain $p(f_*|f_W(\hat{z}_i), \mathcal{D}) = \mathcal{N}(f_*; f_{W_{MAP}}(\hat{z}_i), \phi(y_{z_i,L-1})^T\Lambda\phi(y_{z_i,L-1}))$, where $f_{W_{\mathrm{MAP}}}$ is the output of the neural network at the MAP. In the case of a diagonal Hessian, $\Lambda$ is easily obtained by computing the inverse of each value on the diagonal. For a full Hessian, the process involves first computing the lower triangular decomposition, where $\Lambda^{-1} = L^T L$, then finding the inverse of $L$, and finally obtaining $\Lambda = L^{-1}(L^{-1})^T$.

## B DIAGONAL HESSIAN UNCERTAINTY

Figures 3 and 4 depict the results of DEEPGALA when applying a diagonal approximation to the Hessian to quantify uncertainty for the solutions of PDE1 and PDE2. The mean time to fit and call the diagonal LA for the PDE1 and PDE was 0.07 and 0.004, and 0.067 and 0.06 seconds respectively. As we can see, the uncertainty estimation has been reduced; yet, our method continues to be a good tool to predict how much performance may be improved. However, we still need to understand and quantify how much uncertainty estimation changes when utilising full or diagonal Hessian.

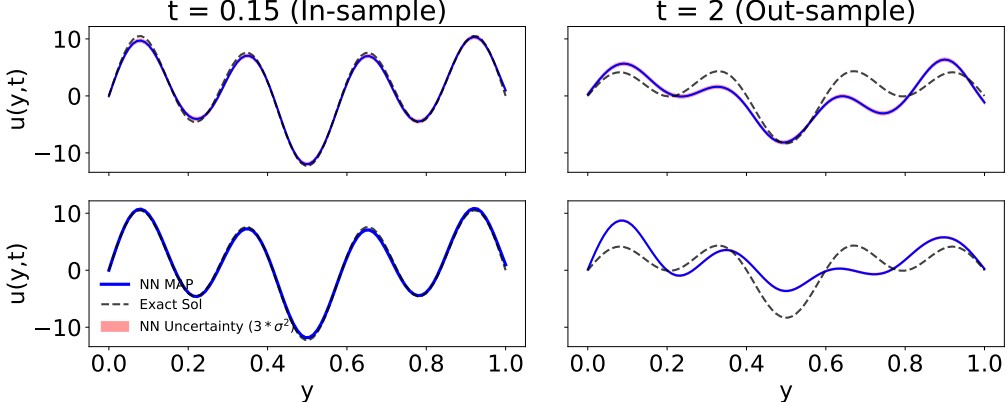

Figure 3: **DEEPGALA Diagonal Hessian Approximation for the PDE1 with $\theta = 0.001$ :** Uncertainty results for in-sample (*left column*) and out-of-sample (*right column*) inputs when DEEPGALA is trained over 1000 (top) and 3000 (bottom) data points respectively.

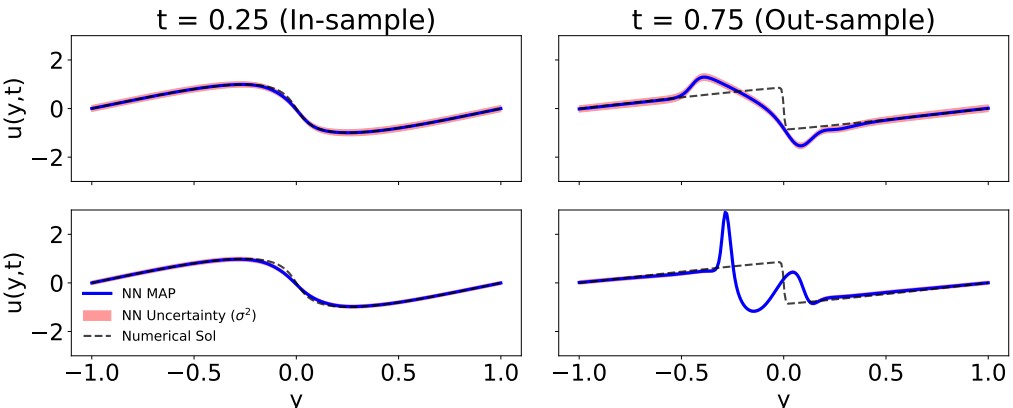

Figure 4: **DEEPGALA Diagonal Hessian Approximation for the PDE2 with** $\theta = 0.01/\pi$ : Uncertainty results for in-sample (*left column*) and out-of-sample (*right column*) inputs when DEEP-GALA is trained over 500 (top) and 1500 (bottom) data points respectively.

