# OpenReview forum: "Galerkin meets Laplace: Fast uncertainty estimation in neural PDEs"
_ICLR.cc/2024/Workshop/AI4DiffEqtnsInSci — AI4DiffEqtnsInSci @ ICLR 2024 Poster_

### Official Review · Reviewer_NpDZ · 2024-02-26
**review of Galerkin meets Laplace: Fast uncertainty estimation in neural PDEs**

**Rating:** 5
**Confidence:** 4

**Review:**

This is a nice paper discussing about uncertainty quantification in physics informed machine learning, by using a Bayesian perspective for the uncertainty quantification of DGM. Here are afew questions:
1. What is the advantage of the propsoed method compared to "Yang, Yibo, and Paris Perdikaris. "Adversarial uncertainty quantification in physics-informed neural networks." Journal of Computational Physics 394 (2019): 136-152."? Why not use the same approach there?
2. Where does the uncertainty comes from? Please be more specific. The authors claimed that "To this end, we first interpret the data D to be the fictious dataset", where does the uncertainty of the collocation points comes from in reality?

Thank you!

---

### Official Review · Reviewer_ocUb · 2024-02-29
**the method makes sense, but the experiments are toy examples**

**Rating:** 6
**Confidence:** 3

**Review:**

Neural PDE solvers need accurate methods for quantifying the approximation uncertainty. This work focuses on the Galerkin method as a neural PDE solver. It then interprets the Galerkin as maximum a posteriori estimator, and then the posterior is approximated via Laplace approximation to capture the uncertainty. Experiments with heat equations and burgers PDE demonstrate the effectiveness of the method.

The proposed method based on Laplace approximation makes sense. The experiments however are for toy PDEs, and more extensive experiments for realistic scenarios are needed to prove the method. The inversion in laplace approximation also impedes scaling the method.

---

### Meta-Review · Area_Chair_WGCb · 2024-02-28

**Recommendation:** Accept (Poster)

**Metareview:**

This paper proposes to add uncertainty quantification, which is very important to the SciML field. to the Deep Galerkin method.  It is nice that the authors consider a shock problem in Burgers' equation and that the uncertainty bounds contain the shock. The authors propose to use a Bayesian approach with the Laplace approximation of the posterior. I would like to see this UQ method compared to other UQ baselines, e.g., simple ensembling, MC-Dropout and the answers to the reviewer's questions addressed. In particular, I agree with (2) to clarify what is meant by "fictious dataset" in the final version and also motivate why the Laplace approximation via the MAP is taken. I vote for acceptance with the above revisions clarified.

---

### Decision · Program_Chairs · 2024-02-28

Accept (Poster)